# Non-Linguistic Supervision for Contrastive Learning of Sentence Embeddings

**Yiren Jian**
Department of Computer Science
Dartmouth College
yiren.jian.gr@dartmouth.edu

**Chongyang Gao**[*]
Department of Computer Science
Northwestern University
chongyanggao2026@u.northwestern.edu

**Soroush Vosoughi**
Department of Computer Science
Dartmouth College
soroush.vosoughi@dartmouth.edu

## Abstract

Semantic representation learning for sentences is an important and well-studied problem in NLP. The current trend for this task involves training a Transformer-based sentence encoder through a contrastive objective with text, i.e., clustering sentences with semantically similar meanings and scattering others. In this work, we find the performance of Transformer models as sentence encoders can be improved by training with multi-modal multi-task losses, using *unpaired* examples from another modality (e.g., sentences and unrelated image/audio data). In particular, besides learning by the contrastive loss on text, our model clusters examples from a non-linguistic domain (e.g., visual/audio) with a similar contrastive loss at the same time. The reliance of our framework on unpaired non-linguistic data makes it language-agnostic, enabling it to be widely applicable beyond English NLP. Experiments on 7 semantic textual similarity benchmarks reveal that models trained with the additional non-linguistic (images/audio) contrastive objective lead to higher quality sentence embeddings. This indicates that Transformer models are able to generalize better by doing a similar task (i.e., clustering) with *unpaired* examples from different modalities in a multi-task fashion. The code is available at https://github.com/yiren-jian/NonLing-CSE.

## 1 Introduction

Learning semantically meaningful sentence embeddings is an important task in Natural Language Processing (NLP), with applications for semantic retrieval and search among other things. The embeddings of sentences are such that semantically similar sentences are located closer to each other in the embedding space. The current state-of-the-art method for learning sentence embeddings is SimCSE [13], which uses contrastive learning. Unsupervised contrastive learning clusters the representations of same examples under different augmentations. SimCSE's main contribution is the use of Dropout augmentation, which they show outperforms other traditional text augmentation methods (e.g., random deletion, swapping, etc.) for learning sentence embeddings. To improve the results further, supervised SimCSE leverages a labeled Natural Language Inference (NLI) dataset during the learning. Each entry of the NLI dataset is a triplet of sentences, a source sentence and its positive and negative pairs (e.g., *"Two dogs are running"* (src), *"There are animals outdoors"* (pos),

---

[*]Contributed as co-first author.

36th Conference on Neural Information Processing Systems (NeurIPS 2022).

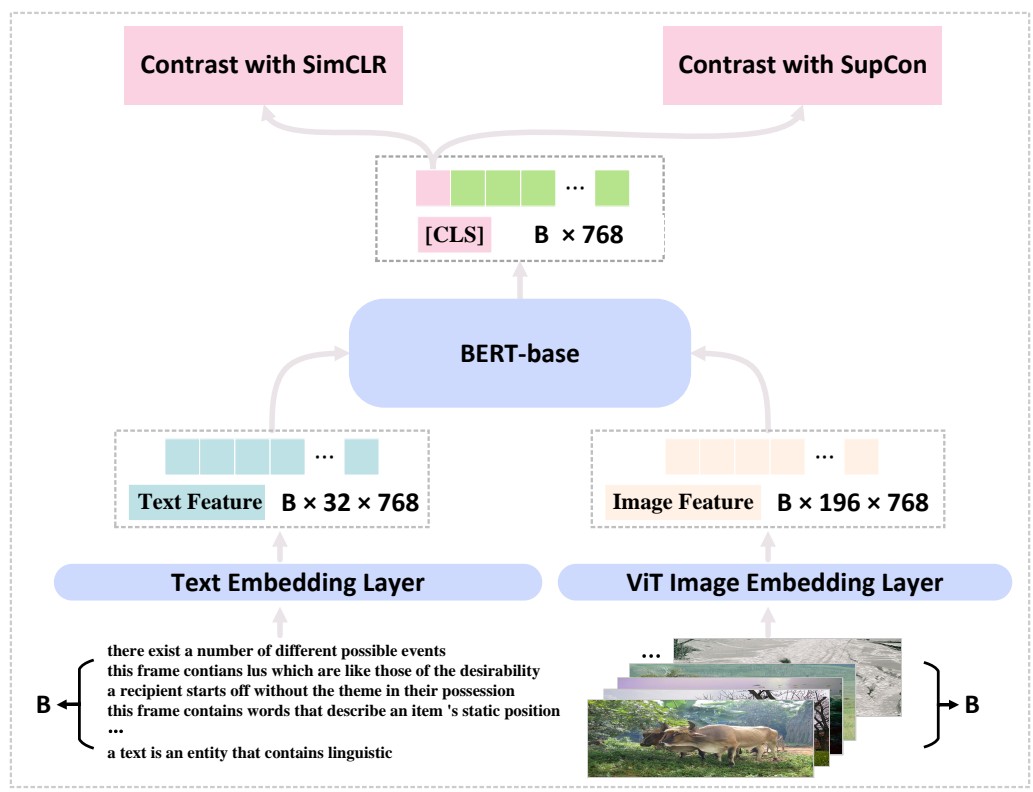

Figure 1: Overview of our method VisualCSE. The mini-batch contains two batches, one with sentences from Wikipedia and the other with images from ImageNet. The text batch is processed by a BERT embedding layer and the image batch is processed by a Vision Transformer (ViT) [11] embedding layer separately. The resulted text and image features are fed into the Transformer encoder and we take the outputs at `[CLS]` position as the representations of examples. Like in SimCSE, the text representations are supervised by the SimCLR loss. We use the SupCon objective to contrast representations from images.

*"The pets are sitting on a couch"* (neg)). Supervised SimCSE uses hard-negative and strong-positive terms in the contrastive learning objective based on the negative and positive examples. Because of the high quality of negative and positive pairs in the NLI dataset, the learning signal from these terms has been shown to be very effective in learning sentence embeddings [13].

However, as we show in this paper, the benefits of a labeled dataset (such as NLI) are highly dependent on its size and the annotation quality. The need for such large and high quality datasets limits the applicability of the method to high-resource languages. For instance, the dataset used by SimCSE is a combination of the SNLI [2] and MNLI [57] datasets which contain 570K and 433K manually annotated examples, respectively. Constructing such a dataset is costly and may be impossible in low-resource languages. In fact, it has been shown that the performance of supervised SimCSE is highly dependent on the specific labeled dataset used. Specifically, training SimCSE with QQP [43], ANLI [37], Flickr30k [62], and ParaNMT [56] was shown to lead to significant performance drops [13]. Such specific reliance on an NLI dataset is especially problematic as constructing it in a new language can be very difficult as it requires annotators to understand each sentence and write a seemingly related, but semantically different sentence. Further supporting this claim, Zhang et al. [64] show that if the distribution of the test set differs significantly from the NLI dataset used for training, the quality of the semantic embeddings is degraded.

The aforementioned issues show that learning sentence embeddings from *unlabeled text data* is a crucial task that requires further exploration. In this paper, we investigate the addition of non-linguistic supervision (provided by easily obtainable public datasets) to contrastive-learning-based sentence

embedding learning. Specifically, we propose a Transformer-based multi-modal multi-task framework which jointly learns two independent contrastive tasks (i.e., one textual and one non-textual) using *unpaired* examples. Our method (shown in Figure 1) is inspired by Lu et al. [35] which shows the ability of Transformers to transfer knowledge from one modality to another (e.g., they show that a Transformer model pre-trained on text can be fine-tuned on downstream visual tasks). In addition to generating higher-quality sentence embeddings, our investigations shed light on the capability of Transformers to learn from mixed modalities when trained on similar tasks. This is complementary to the findings of Lu et al., as those concern the cross-modal transfer ability of Transformers.

Though our method can be used with any modality, we use vision as the modality of choice to describe our method and later show the generalizability of our method to audio. As shown in Figure 1, our method, VisualCSE (or AudioCSE when learning with audio), has a multi-task loss with two terms: one contrastive learning loss for text, which is the same as the one used in SimCSE, and an additional contrastive learning loss to cluster data from images concurrently. Note that the two learning processes are independent of each other, thus allowing us to leverage *unpaired* text and image/audio datasets, e.g., sentences from Wikipedia, images from ImageNet [9], and audio from LibriSpeech [39]. Our method is different than other multi-modal representation learning methods (such as CLIP [44]) in that it does not require paired examples across modalities. This crucial feature makes our method easy to deploy for low-resource languages. In addition, our method also uses a single model with all parameters shared for tasks in both modalities, highlighting the ability of Transformers to learn from non-parallel mixed modalities when trained on similar tasks.

The contributions of this work are as follows:

**(1)** We propose a novel framework for introducing non-linguistic supervision for contrastive-learning-based sentence embedding learning. Crucially, our framework is language-agnostic, does not rely on a specific modality, and does not require paired examples across modalities. We show that the supervision provided by other modalities can improve unsupervised sentence embedding learning, enabling sentence embedding learning in low-resource languages. We improve SimCSE in several semantic textual similarity benchmarks with 3 different pre-trained language models (LMs).

**(2)** Complimentary to the findings of Lu et al. [35], which demonstrate the capability of Transformers to transfer knowledge from language to vision, we show that Transformer models can generalize better by learning a similar task (i.e., clustering) with multi-task losses using non-parallel examples from different modalities.

## 2   Related Work

**Sentence Embedding Learning.** Early works on learning sentence embeddings include training a model to reconstruct the nearby sentences of an encoded sentence [30]. Sent2Vec [38] devises an efficient loss for training the sentence encoder, inspired by the word embedding algorithm Word2Vec. Recent works in this area are based on contrastive learning: through taking two different views of examples [14], data augmentation [59], or by passing examples into different encoders [4, 28, 64]. Post-processing methods also exist for improving the quality of embeddings [23, 32, 47]. The current state-of-the-art sentence embeddings method, SimCSE [13] passes the text into an encoder twice and uses Dropout as the augmentation to create multi-views of examples for contrastive learning. All these methods (and recent works such as DiffCSE [8], DebiasedCSE [65], PromptBERT [25] and many others [3, 16, 22, 26, 33, 42, 45, 51, 52]) solely use text datasets for training their sentence encoders.

A concurrent work, MCSE [63] is the first attempt to learn sentence embedding using text with another modality (image). But MCSE requires *aligned* image-text pairs (i.e., images and captions from Flickr), limiting its applicability to other languages and modalities. We are the first to show that the embedding quality of a sentence encoder can be improved by jointly training with unpaired examples from another modality.

**Contrastive Learning.** Contrastive learning [7, 19, 49, 58, 60] is a form of self-supervised training using a loss derived from noise contrastive estimation [50]. The seminal method SimCLR [5] clusters the same examples under two sets of augmentation (positive pairs), and pulls away different examples in a mini-batch (negative pairs). To alleviate the hardware requirements of SimCLR, MoCo [18] designs a moving average memory bank for storing negative samples, which simulates large number

of negative samples for effective contrastive learning. CMC [49] further extends SimCLR from two-view to multi-view contrastive learning. While the contrastive loss was originally designed for unsupervised pre-training, SupCon [27] finds that additionally contrasting examples based on label information is beneficial when labels of examples are available. In NLP, contrastive learning mainly applies to sentence embedding learning [4, 38, 28, 59, 64], but other works [17, 24] have shown it to be beneficial to few-shot learning [12] as well.

The contrastive objective perfectly matches the goal of sentence embeddings [13], i.e., semantically similar sentences should have comparable representations. Our method for learning sentence embeddings contrasts examples from text and examples from another modality (e.g., image or audio) simultaneously. The image and audio datasets we used come with *"free"* labels, allowing us to apply a loss based on SupCon, in addition to SimCLR.

**Learning Representations With Multiple Modalities.** Using paired image-text instances, CLIP [44] learns aligned representations of text and images by two separate encoders, achieving superior results in several downstream tasks. Building upon CLIP, SLIP [36] further improves the representations of examples by leveraging a self-supervised contrastive objective. UniT [21] learns a multi-modal Transformer model with a set of aligned tasks from visual and language domains. In video learning, XDC [1] learns an encoder by cross pseudo-labeling between parallel RGB frames and audio signals.

The data used for training such models includes images and their associated captions (CLIP and SLIP), aligned visual-language tasks (UniT), or video clips that come with RGB frames and the corresponding audio (XDC). These methods, as well as many other works on multi-modal learning [6, 29, 34, 48, 61], learn better representations by leveraging information shared among different modes of the same instances (i.e., paired examples). Our method also learns better representations of sentences from examples of different modalities. However, the datasets used for training our method do not require parallel examples. Our method performs similar contrastive tasks in two independent domains at the same time, and generalizes better as a sentence encoder by becoming a better clustering model.

## 3 Technical Method

### 3.1 Overview

Given a text dataset $\mathcal{S}_{\text{text}}$ (e.g., Wikipedia or NLI), a dataset from a different modality (e.g., an image dataset $\mathcal{S}_{\text{image}}$ (e.g., ImageNet), or an audio dataset $\mathcal{S}_{\text{audio}}$ (e.g., LibriSpeech)), and a pre-trained LM $\mathcal{M}$ (e.g., BERT), our goal is to fine-tune $\mathcal{M}$ with $\mathcal{S}_{\text{text}}$ and $\mathcal{S}_{\text{image}}/\mathcal{S}_{\text{audio}}$ using contrastive losses such that the resulting model can produce high-quality sentence representations.

In the following sections, we use images as the example to show how non-linguistic supervision can improve the sentence embedding learning (VisualCSE). A similar approach is used to adapt our learning algorithm to audio (AudioCSE), as shown in Appendix B. Our method trains the sentence encoder with a multi-task loss:

$$\mathcal{L}_{\text{total}} = \mathcal{L}_{\text{text}} + \omega_{\text{image}} \mathcal{L}_{\text{image}} \tag{1}$$

where $\omega_{\text{image}}$ is a weighting factor. We provide further details of $\mathcal{L}_{\text{text}}$ and $\mathcal{L}_{\text{image}}$ in Sections 3.2 and 3.3, respectively.

### 3.2 Learning from Text

We train the sentence encoder with text using the method laid out by SimCSE. This allows us to directly investigate the contribution of additional non-linguistic supervision for training sentence encoders. Here, we briefly summarize how SimCSE works.

SimCSE learns the sentence encoder by a contrastive loss, using Dropout as a form of data augmentation. For a single unlabeled example $x_i$ of $\mathcal{S}_{\text{text}}$ (Wikipedia) in a mini-batch, SimCSE passes $x_i$ into the Transformer encoder $\mathcal{M}$ twice. The corresponding output hidden states at [CLS] position (which are used as sentence embeddings) are denoted as $\mathbf{h}_i^{z_i}$ and $\mathbf{h}_i^{z_i'}$, where $z_i$ and $z_i'$ are two different random masks by Dropout layers of the Transformer model. SimCSE applies the SimCLR loss to contrast examples in a mini-batch, i.e., the unsupervised loss based on data augmentation is as

follows:

$$\mathcal{L}_{\text{text}}^{\text{unsup}} = \sum_{i=1}^{N} -\log \frac{e^{\text{sim}(\mathbf{h}_i^{z_i}, \mathbf{h}_i^{z_i'})/\tau}}{\sum_{j=1}^{N} e^{\text{sim}(\mathbf{h}_i^{z_i}, \mathbf{h}_j^{z_j'})/\tau}} \tag{2}$$

where $j$ denotes another example in the mini-batch, $N$ is the size of mini-batch, sim is the operation for computing cosine similarity, and $\tau$ is a temperature scaling factor. The key to SimCSE is using Dropout as a form of augmentation, the rest of its contrastive learning follows SimCLR.

If $\mathcal{S}_{\text{text}}$ is a labeled NLI dataset that comes with $(x_i, x_i^+, x_i^-)$ (i.e., source, entailment, contradiction), corresponding to the hidden states $(h_i, h_i^+, h_i^-)$, its supervised loss is defined as:

$$\mathcal{L}_{\text{text}}^{\text{sup}} = \sum_{i=1}^{N} -\log \frac{e^{\text{sim}(\mathbf{h}_i, \mathbf{h}_i^+)/\tau}}{\sum_{j=1}^{N} (e^{\text{sim}(\mathbf{h}_i, \mathbf{h}_j^+)/\tau} + e^{\text{sim}(\mathbf{h}_i, \mathbf{h}_j^-)/\tau})} \tag{3}$$

Following SimCSE, we use $\tau = 0.05$ for both losses.

## 3.3 Learning from Images

We train our encoder with images using SupCon (main results) and SimCLR (in ablation). For an image $y_i$ in $\mathcal{S}_{\text{image}}$, we apply two sets of data augmentations in the input RGB space. The resulting views of the input $(y_i', y_i'')$ are passed into a ViT embedding layer and further encoded by the same Transformer encoder $\mathcal{M}$ from Section 3.2. The SupCon contrastive loss is applied on the two output representations $(\mathbf{f}_i', \mathbf{f}_i'')$ at the [CLS] position:

$$\mathcal{L}_{\text{image}}^{\text{SupCon}} = \sum_{i=1}^{N} -\log \frac{e^{\text{sim}(\mathbf{f}_i', \mathbf{f}_i'')/\tau} + \sum_{y_i \text{ and } y_j \text{ from same class}} e^{\text{sim}(\mathbf{f}_i', \mathbf{f}_j'')/\tau}}{\sum_{y_i \text{ and } y_j \text{ from different class}} e^{\text{sim}(\mathbf{f}_i', \mathbf{f}_j'')/\tau}} \tag{4}$$

We use the SupCon default $\tau = 0.07$. Also, we can substitute SupCon loss with the SimCLR:

$$\mathcal{L}_{\text{image}}^{\text{SimCLR}} = \sum_{i=1}^{N} -\log \frac{e^{\text{sim}(\mathbf{f}_i', \mathbf{f}_i'')/\tau}}{\sum_{j=1}^{N} e^{\text{sim}(\mathbf{f}_i', \mathbf{f}_j'')/\tau}} \tag{5}$$

As we show in Section 4.4, SimCLR and SupCon produce in similar performances. This is because we use a very small batch size (48) due to the limitations of our GPU memory. Thus, the second term in the numerator of $\mathcal{L}_{\text{text}}^{\text{SupCon}}$ is nearly non-existent, and Equation 4 reduces to Equation 5.

Note that $\mathcal{L}_{\text{image}}$ does not make any assumptions on the image dataset. Thus, we can leverage the easily obtainable ImageNet images. Furthermore, $\mathcal{L}_{\text{image}}^{\text{SimCLR}}$ can be applied to any unlabeled images from the Internet, at no cost. This allows our framework to be easily deployed for learning sentence embeddings in a new domain or a new language where NLI, or other high-quality labeled datasets are not available.

## 3.4 PyTorch Style of Pseudo-Code

We provide the pseudo-code of our algorithm VisualCSE in the style of PyTorch in Algorithm 1. Learning AudioCSE is similar except that we do not apply transform() to the audio data, but instead pass the audio data into the model twice (i.e., augmentation is provided through Dropout).

## 3.5 Why Does It Work?

As we can see, $\mathcal{L}_{\text{text}}$ and $\mathcal{L}_{\text{image}}$ share a similar (if not the same) form, with the only difference being the input data. We speculate that the Transformer model becomes a better contrastive learner by jointly training with similar but independent contrastive objectives in different modalities.

**Algorithm 1:** Visual contrastive sentence embedding learning (VisualCSE)

```
# Max_Step:         Number of training steps
# LM:               The language model
# D_text:           Dataloader for text training set
# D_image:          Dataloader for image training set
# transform:        A set of data transformations
# SimCLR:           SimCLR loss
# SupCon:           SupCon loss
# opt_text:         Optimizer for learning from text
# opt_image:        Optimizer for learning from image
for i in Max_Step:
    ### Retrieve two augmented views of a same sent/img
    sent = next(D_text)
    sent_0, sent_1 = sent, sent
    img, y = next(D_image)
    img_0, img_1, = transform(img), transform(img)
    ### Learning from text
    L_text = SimCLR(LM(sent_0), LM(sent_1))
    L_text.backward()
    opt_text.step()
    ### Learning from images
    L_img = SupCon(LM(img_0), LM(img_1), y)
    L_img.backward()
    opt_image.step()
return LM
```

# 4 Experiments

## 4.1 Setup

**Evaluation.** Following standard practice for sentence embedding, we evaluate our approach on 7 semantic textual similarity (STS) tasks: STS 2012-2016, STS-Benchmark and SICK-Relatedness. For fair comparison, we use the same evaluation protocol as SimCSE [13], and use Spearman's correlation as the metric.

**Models and Training Datasets.** We test our method with three different pre-trained Transformer models used in SimCSE: BERT-base-uncased, RoBERTa-base and RoBERTa-large. We fine-tune the pre-trained models with our learning objective, shown in Eq. 1.

For learning with $\mathcal{L}_{\text{text}}$, we use $10^6$ sentences down-sampled from the Wikipedia English dataset for unsupervised sentence embedding learning (Eq. 2). For supervised sentence embedding learning (Eq. 3), we (and SimCSE) use a combined NLI dataset with 314K sentences with paired examples labeled as entailment, neutral, and non-entailment. SimCSE uses entailment examples as strong-positives and non-entailment examples as hard-negatives.

For learning with $\mathcal{L}_{\text{image}}$, both unsupervised and supervised sentence embedding settings use a down-sampled ImageNet dataset $\mathcal{S}_{\text{image}}$. Images from $\mathcal{S}_{\text{image}}$ are chosen *randomly* from 60 classes with each class having 500 images[2]. The images have a shape of $B \times 3 \times 224 \times 224$, thus the image features processed by the ViT embedding layers are tensors of size $B \times 197 \times h$, where $B$ is the batch size and $h$ is the hidden dimension of the LM. Learning with audio is described in Appendix B.

**Optimization Details.** Following SimCSE [13], we train unsupervised models with AdamW for one epoch, and supervised models for 3 epochs. We then search batch sizes and learning rates from $\{64, 128, 256\}$ and $\{1e^{-5}, 2e^{-5}, 3e^{-5}\}$ for $\mathcal{L}_{\text{text}}$. Moreover, we use a fixed batch size of 48 for $\mathcal{L}_{\text{image}}$ (and $\mathcal{L}_{\text{audio}}$) and search learning rates among $\{5e^{-6}, 2e^{-6}1e^{-6}, 5e^{-7}, 2e^{-7}, 1e^{-7}\}$. The models are selected based on the validation set of the STS-Benchmark.

All the unsupervised base LMs are trained on 24GB Nvidia RTX-6000 GPUs, while supervised and large models are trained on 48GB Nvidia RTX-A6000 GPUs. We use pytorch-1.10 with CUDA 11.3,

---

[2]It takes SimCSE and VisualCSE thousands of steps to converge. Thus, we construct a smaller subset of ImageNet so that the model goes over the dataset for a few epochs.

torchvision-0.11.3, torchaudio-0.10.2, and Huggingface transformers-4.5.0 for our implementation. Training with a different version of software and hardware may lead to different results. We use the default random seed 42 (same as SimCSE) of Huggingface for all of our major experiments for reproducibility.

The validated hyper-parameters can be found in Appendix A.

## 4.2 Main Results

We show results of our method learning with visual supervision (VisualCSE) in the unsupervised text setting in Table 1. Our method outperforms baseline SimCSE that learns only from unlabeled text with three different pre-trained LMs. Similarly, we show in Table 2 that learning with audio supervision can improve over SimCSE in almost all tasks.

The improvement in performance can be attributed to the fact that SimCSE only clusters text examples, while VisusalCSE or AudioCSE simultaneously cluster text and examples from another modality, making them better contrastive learners. This could indicate that Transformer-based models are able to generalize better by doing the same task (e.g., clustering) with independent examples from different modalities in a multi-task fashion (i.e, with a multi-task loss). Importantly, this learning setup does not require any paired examples across modalities.

| Model | STS12 | STS13 | STS14 | STS15 | STS16 | STS-B | SICK-R | Avg. |
|---|---|---|---|---|---|---|---|---|
| *Unsupervised models* | | | | | | | | |
| SimCSE-BERT$_{base}$ ♠ | 68.40 | 82.41 | 74.38 | 80.91 | 78.56 | 76.85 | **72.23** | 76.25 |
| VisualCSE-BERT$_{base}$ | **71.16** | **83.29** | **75.13** | **81.59** | **80.05** | **80.03** | 71.23 | **77.50** |
| SimCSE-RoBERTa$_{base}$ ♠ | 70.16 | 81.77 | 73.24 | 81.36 | 80.65 | 80.22 | 68.56 | 76.57 |
| VisualCSE-RoBERTa$_{base}$ | **70.41** | **83.51** | **74.87** | **82.79** | **81.67** | **81.89** | **69.95** | **77.87** |
| SimCSE-RoBERTa$_{large}$ ♠ | 72.86 | 83.99 | 75.62 | 84.77 | 81.80 | 81.98 | 71.26 | 78.90 |
| VisualCSE-RoBERTa$_{large}$ | **73.09** | **84.77** | **77.09** | **85.47** | **82.06** | **83.26** | **72.23** | **79.71** |

Table 1: Spearman's correlation of different sentence embedding methods using unsupervised text. Comparing to SimCSE that only leverages text, our VisualCSE takes addition supervision from *images*. ♠: results from the SimCSE paper and reproduced by officially released models.

| Model | STS12 | STS13 | STS14 | STS15 | STS16 | STS-B | SICK-R | Avg. |
|---|---|---|---|---|---|---|---|---|
| *Unsupervised models* | | | | | | | | |
| SimCSE-BERT$_{base}$ ♠ | 68.40 | 82.41 | 74.38 | 80.91 | 78.56 | 76.85 | **72.23** | 76.25 |
| AudioCSE-BERT$_{base}$ | **71.65** | **84.27** | **76.69** | **83.22** | **78.69** | **79.94** | 70.49 | **77.85** |
| SimCSE-RoBERTa$_{base}$ ♠ | **70.16** | 81.77 | 73.24 | 81.36 | 80.65 | 80.22 | 68.56 | 76.57 |
| AudioCSE-RoBERTa$_{base}$ | 68.44 | **83.96** | **75.77** | **82.38** | **82.07** | **81.63** | **70.56** | **77.83** |
| SimCSE-RoBERTa$_{large}$ ♠ | **72.86** | 83.99 | 75.62 | 84.77 | 81.80 | 81.98 | 71.26 | 78.90 |
| AudioCSE-RoBERTa$_{large}$ | 72.10 | **84.30** | **76.74** | **85.11** | **82.51** | **82.94** | **72.45** | **79.45** |

Table 2: Spearman's correlation of different sentence embedding methods using unsupervised text. Comparing to SimCSE that only leverages text, our AudioCSE takes addition supervision from *audio*.

Though not being the main goal of sentence embedding, we evaluate our models on transfer tasks in Appendix D, showing that the sentence representations learned using our framework can be well-suited to downstream tasks. We also show example sentence retrieval experiments in Appendix C and VisualCSE with lower quality CIFAR images in Appendix F.

## 4.3 Repeated Experiments

SimCSE uses a fixed random seed (42) for training their models, selects models using the STS-B validation set and reports results on all 7 testing sets of the benchmarks. For direct comparison, we follow this setup in Section 4.2. To investigate the statistical significance of our results, we further run repeated experiments for unsupervised VisualCSE and SimCSE using 5 different random seeds. Because the best validated model is saved at one point during the whole learning process, using different random seeds for training VisualCSE/SimCSE can be thought of as simulating scenarios

where different subsets of the text and image datasets are used for training the models. As shown in Table 3, we see a performance drop for both SimCSE and VisualCSE using BERT-base. However, VisualCSE, using either model, is shown to be statistically significantly better than SimCSE.

## 4.4 Ablation on $\mathcal{L}_{\text{image}}$

By default we use a variant of SupCon (Eq. 4) as our contrastive objective for images when learning with visual supervision. This allows us to leverage the label information available in the ImageNet dataset. Besides clustering different views of examples in a mini-bacth, SupCon also tries to pull together examples in a mini-batch that share the same label.

| Model | Avg. | $p$ |
|---|---|---|
| SimCSE-BERT$_{\text{base}}$ | $75.74_{\pm 0.90}$ | 0.0235 |
| VisualCSE-BERT$_{\text{base}}$ | $\mathbf{76.96}_{\pm 0.38}$ | |
| SimCSE-RoBERTa$_{\text{base}}$ | $76.36_{\pm 0.18}$ | <0.0001 |
| VisualCSE-RoBERTa$_{\text{base}}$ | $\mathbf{77.74}_{\pm 0.25}$ | |

Table 3: 5 repeated experiments of unsupervised experiments. We report means and standard deviations for average Spearman's correlation of the 7 STS benchmarks.

However, in practice, due to the limitations of our GPU memory, we train VisualCSE with a batch size of 48. As we have 60 image classes, each with 500 images, in our down-size ImageNet dataset used for training, it is not likely that multiple examples from the same class are contained in one mini-batch. Thus, the SupCon loss in practice degenerates to the SimCLR loss. As we can see from Table 4, changing SupCon to SimCLR does not greatly affect the performance (ablation on $\mathcal{L}_{\text{audio}}$ shows similar trends as shown in Appendix E).

The results with SimCLR show that our framework can be successfully deployed even when only unlabeled non-linguistic datasets are available. Nevertheless, in this paper, since we use labeled image and audio datasets, we use SupCon as our default loss to possibly harvest the label information presented in our datasets (though this is unlikely given our batch size).

| | STS12 | STS13 | STS14 | STS15 | STS16 | STS-B | SICK-R | Avg. |
|---|---|---|---|---|---|---|---|---|
| BERT (SimCLR) | 71.81 | 82.57 | 75.40 | 82.58 | 79.34 | 78.96 | 70.83 | 77.36 |
| BERT (SupCon) | 71.16 | 83.29 | 75.13 | 81.59 | 80.05 | 80.03 | 71.23 | 77.50 |
| RoBERTa (SimCLR) | 71.02 | 83.19 | 74.47 | 82.59 | 81.95 | 81.94 | 69.87 | 77.86 |
| RoBERTa (SupCon) | 70.41 | 83.51 | 74.87 | 82.79 | 81.67 | 81.89 | 69.95 | 77.87 |

Table 4: Comparison of VisualCSE (base) using SupCon and SimCLR as $\mathcal{L}_{\text{image}}$.

## 4.5 VisualCSE on Different Languages

One of the main advantages of our proposed framework is its ability to generate higher quality sentence embeddings than unsupervised SimCSE, without the need for additional labeled datasets (as opposed to supervised SimCSE, which requires a large, high-quality, and difficult to annotate NLI dataset, see Section 4.6). This is especially important in low-resource languages where NLI datasets are hard to acquire. Here, we show the generalizability of our framework to other languages. Specifically, we experiment on German, French and Russian, using BERT-base models from HuggingFace (see Appendix G). We follow the procedure in SimCSE to construct 1 million sentences from Wikipedia of German, French, Russian and Chinese as training datasets. The evaluations are done on validation sets of the multilingual STS-B from HuggingFace. As shown in Table 5,

| Language | Model | Spearman's |
|---|---|---|
| German | SimCSE | 67.34 |
| | VisualCSE | **69.87** |
| Chinese | SimCSE | 67.98 |
| | VisualCSE | **70.05** |
| French | SimCSE | 70.31 |
| | VisualCSE | **72.52** |
| Russian | SimCSE | 72.50 |
| | VisualCSE | **77.48** |

Table 5: SimCSE and VisualCSE with different languages.

VisualCSE can improve SimCSE in all three languages by leveraging additional visual supervision, with Russian seeing an improvement of 4.98 in the Spearman's correlation.

## 4.6 Supervised Sentence Embedding

Lastly, we investigate whether supervision from other modalities can improve *supervised* sentence embedding. In this setting, the text contrastive loss, $\mathcal{L}_{\text{text}}$, uses entailment pairs as strong positives in the numerator and has an extra term in the denominator as the hard-negative learning signal to push away non-entailment pairs in the NLI dataset (see Eq. 3). Since in our image dataset $\mathcal{S}_{\text{image}}$ (or audio dataset $\mathcal{S}_{\text{audio}}$) we do not have such labeled examples to form the hard-negative pairs, we apply the same unmodified $\mathcal{L}_{\text{image}}$ (or $\mathcal{L}_{\text{audio}}$) to the supervised sentence embedding learning. In Table 6, we observe that Visual/AudioCSE can still outperform supervised SimCSE in all the tasks, but with much smaller margins. This shows that the learning signal from the high-quality negative and positive pairs of the NLI dataset are very strong (leading to a 5.32 improvement over unsupervised SimCSE) and cannot be supplemented by supervision from other modalities.

| Model | STS12 | STS13 | STS14 | STS15 | STS16 | STS-B | SICK-R | Avg. |
|---|---|---|---|---|---|---|---|---|
| *Supervised models* | | | | | | | | |
| SimCSE-BERT$_{\text{base}}$ ♠ | 75.30 | 84.67 | 80.19 | 85.40 | 80.82 | 84.25 | 80.39 | 81.57 |
| VisualCSE-BERT$_{\text{base}}$ | 75.39 | 84.69 | **80.34** | 85.76 | **81.60** | **84.85** | 80.39 | **81.86** |
| AudioCSE-BERT$_{\text{base}}$ | **75.73** | **84.92** | 80.25 | **85.90** | 81.00 | 84.36 | **80.75** | 81.84 |

Table 6: Spearman's of different sentence embedding learning methods in the supervised setting.

# 5 Analysis

## 5.1 Visual Supervision Versus NLI Supervision

Given the strong performance of supervised SimCSE with the NLI dataset, natural questions to ask are: (1) How much NLI data can be compensated for through visual supervision of unsupervised VisualCSE? (2) How sensitive is supervised SimCSE to the quality of the labeled dataset used for the hard negatives? Or in other words, at what level of noise in the NLI dataset, does the unsupervised VisualCSE's performance surpass that of supervised SimCSE?

To answer the first question, we down-size the NLI dataset used in supervised SimCSE into different sizes and train SimCSE with these sub-NLI datasets. As shown in Figure 2, our unsupervised VisualCSE works roughly equal to supervised SimCSE with 30K NLI examples.

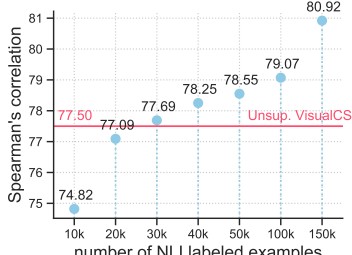

Figure 2: Supervised SimCSE with different subsets of the NLI dataset.

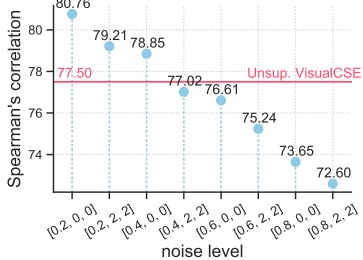

Figure 3: Supervised SimCSE with different levels of noise. E.g., [0.3,2,2] is applying 30% random deletion, and 2 insertions and swaps.

As Gao et al. show, the performance of supervised SimCSE is sensitive to the dataset used for supervision in Eq. 3 [13]. For instance, they show that replacing the NLI dataset with ParaNMT can reduce the performance by 6 points (i.e., degrading to the performance of unsupervised SimCSE). Furthermore, they show that using only contradiction without entailment of the NLI also drops the performance by 7 points (they also show performance drops with other datasets, such as QQP, Flickr30k, and ANLI). This indicates that the performance of contrastive-learning-based supervised models for sentence embedding relies heavily on the high quality of the NLI dataset.

To explore this, we further run experiments on supervised SimCSE with a noisy NLI dataset (at different levels of noise) and observe the point at which our unsupervised VisualCSE starts outperforming

supervised SimCSE. We add noise through deletion, insertion, and swapping. As shown in Figure 3, the performance of supervised SimCSE clearly goes down as the quality of the data is degraded, going below unsupervised VisualCSE at around $40\%$ random deletions.

## 5.2 Uniformity and Alignment of the Sentence Embeddings

The quality of learned representations can also be measured by *Uniformity* and *Alignment* [53]. *Alignment* is computed based on the expected distance between representations of positive pairs. Lower *Alignment* indicates close representations between positive pairs: $\text{Align} = \mathbb{E}_{(x,x^+) \sim p_{\text{pos}}} |f(x) - f(x^+)|^2$. *Uniformity*, on the other hand, measures how well representations are uniformly scattered across the representation space: $\text{Uniform} = \mathbb{E}_{(x,y) \sim p_{\text{data}}} e^{-2|f(x) - f(y)|^2}$.

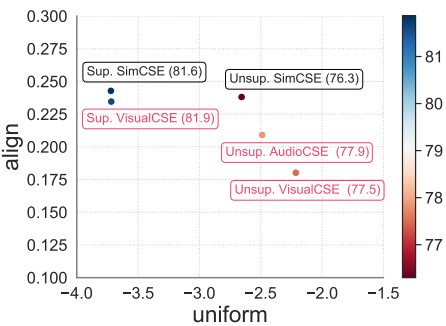

Figure 4: *Alignment* v.s. *Uniformity* of models based on BERT$_{\text{base}}$ (ours in red).

We show the *Uniformity* and *Alignment* plot of our models evaluated on the STS-B in Figure 4. Our VisualCSE and AudioCSE exhibit better *Alignment* than the text-only SimCSE models, both in supervised and unsupervised settings. However, note that the unsupervised Visual/AudioCSE suffer in *Uniformity*, this is especially true of VisualCSE, which though has better *Alignment* than AudioCSE, has a lower performance due to its worse *Uniformity*. These findings indicate that the non-linguistic supervision in our models make them better clusterers (which is basically what *Alignment* measures).

## 6 Discussion and Limitations

We have shown that non-linguistic supervision provided by unpaired data from the visual and audio domain can improve the performance of contrastive-learning-based sentence embedding learners in *unsupervised* settings. Though this extra supervision does not fully compensate for the presence of the large high-quality labeled dataset used in *supervised* sentence embedding, we show that it can indeed compensate for smaller, noisier versions. Our findings are important from both a technical point of view and their implications for broader impacts. Our findings indicate that Transformer models can generalize better as contrastive-learners by learning to cluster with *independent* data from different modalities. Future work could investigate whether this behavior is true for tasks other than clustering (e.g., classification). In terms of broader impact, as we have shown, our framework is language-agnostic, enabling it to be deployed for low-resource languages where large, high-quality, and hard-annotated labeled datasets may be hard to obtain. This is possible because our method does not require *paired* datasets (i.e., the same image/audio dataset can be used for any language).

A limitation of our method is the reduction in *Uniformity* of the sentence embeddings. This means that the representations learned may be grouped together in regions of the representation space. Future work could investigate the reason behind this phenomenon and propose methods to alleviate it. Another limitation of our method is the increase in the training time, as it takes twice as long to train our models since we require an additional forward/backward pass for the non-linguistic data.

## 7 Conclusion

In this paper, we find that sentence representation learning can be improved by contrasting *unpaired* examples from non-linguistics domains, in addition to text. Our framework for incorporating such non-linguistic supervision for sentence embedding is language- and modality-agnostic. Specifically, we show that both visual and audio supervision from unpaired examples can improve unsupervised (and to a lesser extent, supervised) contrastive-learning based sentence representation learners across various languages. Our models, Visual/AudioCSE achieve superior performances in standard sentence embedding benchmarks. Due to its reliance on unpaired data from non-linguistic modalities and its language-agnostic nature, our framework has the potential to be used for low-resource languages.

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
