# Non-Linguistic Supervision for Contrastive Learning of Sentence Embeddings

# Appendix

**Yiren Jian**
Department of Computer Science
Dartmouth College
yiren.jian.gr@dartmouth.edu

**Chongyang Gao**[*]
Department of Computer Science
Northwestern University
chongyanggao2026@u.northwestern.edu

**Soroush Vosoughi**
Department of Computer Science
Dartmouth College
soroush.vosoughi@dartmouth.edu

## A   Training Details

We provide hyper-parameters of our models in Table A.1. During the training, our models are evaluated on the STS-B validation set and best models are finally tested on testing sets of the 7 STS tasks.

| Model | GPU | Text Bsz | Text LR | Img/Audio Bsz | Img/Audio LR |
|---|---|---|---|---|---|
| VisualCSE-BERT-base | RTX-6000 | 64 | $3e^{-5}$ | 48 | $1e^{-6}$ |
| AudioCSE-BERT-base | RTX-6000 | 64 | $3e^{-5}$ | 48 | $2e^{-7}$ |
| VisualCSE-RoBERTa-base | RTX-6000 | 128 | $2e^{-5}$ | 48 | $5e^{-7}$ |
| AudioCSE-RoBERTa-base | RTX-6000 | 128 | $2e^{-5}$ | 48 | $2e^{-7}$ |
| VisualCSE-RoBERTa-large | RTX-A6000 | 256 | $1e^{-5}$ | 48 | $5e^{-7}$ |
| AudioCSE-RoBERTa-large | RTX-A6000 | 256 | $1e^{-5}$ | 48 | $5e^{-6}$ |

Table A.1: Hyper-parameters used for training our VisualCSE and AudioCSE. All the batch sizes (Bsz) and learning rates (LR) are selected based on validation on STS-B.

## B   Learning Sentence Embeddings with Audio Supervision

Learning AudioCSE using audio is very similar to VisualCSE, with the only difference being in pre-processing of the raw data. The raw audio data has the shape $B \times 1024 \times 128$. We use publicly available code from Gong et al. [2] (called SSAST) to process the data so that each example is represented by a tensor of shape $1 \times 211 \times h$. This tensor is then passed into the Transformer encoder during the training.

Different from image learning that allows us to leverage common data augmentations in Computer Vision, we use Dropout augmentation (the same strategy in SimCSE) for AudioCSE. For an audio example, the two views of the input $(y_i', y_i'')$ are passed into the embedding layer of SSAST. The

---

[*]Contributed as co-first author.

36th Conference on Neural Information Processing Systems (NeurIPS 2022).

SupCon contrastive loss is applied at the two output representations $(\mathbf{f}'_i, \mathbf{f}''_i)$ at the `[CLS]` position:

$$\mathcal{L}_{\text{audio}}^{\text{SupCon}} = \sum_{i=1}^{N} - \log \frac{e^{\text{sim}(\mathbf{f}'_i, \mathbf{f}''_i)/\tau} + \sum\limits_{y_i \text{ and } y_j \text{ from same class}} e^{\text{sim}(\mathbf{f}'_i, \mathbf{f}''_j)/\tau}}{\sum\limits_{y_i \text{ and } y_j \text{ from different class}} e^{\text{sim}(\mathbf{f}'_i, \mathbf{f}''_j)/\tau}} \tag{1}$$

Or, the SimCLR loss equivalent:

$$\mathcal{L}_{\text{audio}}^{\text{SimCLR}} = \sum_{i=1}^{N} - \log \frac{e^{\text{sim}(\mathbf{f}'_i, \mathbf{f}''_i)/\tau}}{\sum_{j=1}^{N} e^{\text{sim}(\mathbf{f}'_i, \mathbf{f}''_j)/\tau}} \tag{2}$$

Same as with images, the audio dataset is not paired with the text data and $\mathcal{L}_{\text{audio}}$ does not make any assumption or have any constraints on the audio data that is used for training, meaning that datasets other than LibriSpeech can be used by our framework as well.

For learning with $\mathcal{L}_{\text{audio}}$, we use `train-clean-100` split from the LibriSpeech dataset [5]. We use a subset of LibriSpeech $\mathcal{S}_{\text{audio}}$ comprising of 29K examples from 50 classes.

## C  Qualitative Analysis

We compare unsup-SimCSE and unsup-VisualCSE on a small scale retrieval test. We use 150K captions in Flicker30k as the database and retrieve the top-3 most similar sentences of a query by the two models. As shown in Table C.1, VisualCSE generally retrieves qualitatively different sentences than SimCSE.

|  | unsup-SimCSE-BERT-base | unsup-VisualCSE-BERT-base |
|---|---|---|
| **Query**: A double decker red bus moving along the street. | | |
| #1 | A bus with people on it. | A yellow van driving down crowded city street. |
| #2 | A large orange bus is stopped on the street. | A red double-decker bus in Europe. |
| #3 | The blue bus passes pedestrians on a busy city street. | A city street is shown with people and double-decker buses. |
| **Query**: A couples posts for a picture in front of a bus. | | |
| #1 | A couple are paying their fare at the front of a bus. | A couple are paying their fare at the front of a bus. |
| #2 | Two women are walking in a crosswalk with a bus ... | A young couple riding on a bus with the boy 's arm around the girl. |
| #3 | A man and woman pose in front of some traffic. | A man and a woman sit on a bus , but not together. |
| **Query**: A cat sits inside of a paper airplane toy. | | |
| #1 | A cat is watching a girl construct a Lego airplane. | A cat is watching ... a young girl playing with a toy airplane. |
| #2 | A small child plays with her airplane as a cat looks on. | A small child plays with her airplane as a cat looks on. |
| #3 | A house cat is sitting on top of an old white car. | A cat is watching a girl construct a Lego airplane. |

Table C.1: Examples of retrieved top three examples by SimCSE and VisualCSE from the Flickr30k dataset (150k sentences) using random queries.

## D  Transfer Tasks

Though downstream classification tasks are not the main goal of sentence representation learning, we still evaluate VisualCSE/AudioCSE in the following transfer tasks: MR [7], CR [3], SUBJ [6], MPQA [10], SST-2 [8], TREC [9], and MRPC [1] using Facebook's publicly available code base of SentEval. Following default settings from SentEval, a linear classifier is trained on top of the frozen sentence embeddings of different models, i.e., SimCSE and VisualCSE.

Table D.1 shows that VisualCSE and AudioCSE generally outperfom SimCSE in the transfer benchmarks, though some improvements are marginal. These findings show that the representations learned by our framework can be successfully applied to downstream tasks.

## E  Ablation on $\mathcal{L}_{\text{audio}}$

Following what was done in Section 4.4, we study AudioCSE with two different contrastive losses (SimCLR and SupCon). The results in Table E.1 show a similar trend to the results for VisualCSE, shown in Table 4, i.e., changing SupCon to SimCLR does not greatly affect the performance.

| Model | MR | CR | SUBJ | MPQA | SST | TREC | MRPC | Avg. |
|---|---|---|---|---|---|---|---|---|
| *Unsupervised models* | | | | | | | | |
| SimCSE-BERT$_{base}$ ♠ | 81.18 | 86.46 | 94.43 | 88.87 | 85.50 | 89.80 | **74.49** | 85.82 |
| VisualCSE-BERT$_{base}$ | **82.29** | **87.18** | **94.85** | **89.34** | **86.55** | 89.60 | 74.32 | **86.30** |
| AudioCSE-BERT$_{base}$ | 81.10 | 86.76 | 94.65 | 88.78 | 86.00 | 89.40 | 74.43 | 85.87 |
| SimCSE-RoBERTa$_{base}$ ♠ | 81.04 | **87.74** | 93.28 | 86.94 | 86.60 | 84.60 | 73.68 | 84.84 |
| VisualCSE-RoBERTa$_{base}$ | **81.84** | 87.60 | 92.76 | 87.14 | 86.77 | 85.20 | 74.43 | 85.11 |
| AudioCSE-RoBERTa$_{base}$ | 80.76 | 87.18 | **93.39** | **87.29** | **87.64** | **86.60** | **75.07** | **85.42** |
| SimCSE-RoBERTa$_{large}$ ♠ | 82.74 | 87.87 | 93.66 | 88.22 | 88.58 | 92.00 | 69.68 | 86.11 |
| VisualCSE-RoBERTa$_{large}$ | 83.44 | 88.13 | **93.96** | 88.43 | **88.63** | 91.60 | **73.86** | **86.86** |
| AudioCSE-RoBERTa$_{large}$ | **83.49** | **88.64** | 93.86 | **88.82** | **88.63** | 92.40 | 69.97 | 86.54 |

Table D.1: Testing accuracy on transfer tasks using different models. ♠: results from SimCSE's publicly available model.

| | STS12 | STS13 | STS14 | STS15 | STS16 | STS-B | SICK-R | Avg. |
|---|---|---|---|---|---|---|---|---|
| BERT (SimCLR) | 70.98 | 83.03 | 75.74 | 83.03 | 78.10 | 78.33 | 69.74 | 77.00 |
| BERT (SupCon) | 71.65 | 84.27 | 76.69 | 83.22 | 78.69 | 79.94 | 70.49 | 77.85 |
| RoBERTa (SimCLR) | 68.88 | 83.74 | 74.69 | 82.46 | 82.02 | 81.52 | 70.65 | 77.71 |
| RoBERTa (SupCon) | 68.44 | 83.96 | 75.77 | 82.38 | 82.07 | 81.63 | 70.56 | 77.83 |

Table E.1: Comparison of AudioCSE (base) using SupCon and SimCLR as $\mathcal{L}_{image}$.

# F  VisualCSE with CIFAR images

We further carry out experiments on substituting ImageNet with a lower quality CIFAR [4] dataset for VisualCSE. CIFAR images have a shape of 32x32 and we intentionally resize (enlarge) them to 224x224 to be encoded by the ViT embedding layer. This interpolation causes the CIFAR images to become blurry and lower quality. As shown in Table F.1 our framework improves over SimCSE even with this lower quality dataset (results shown below).

| Model | Avg. |
|---|---|
| SimCSE-BERT$_{base}$ | 76.25 |
| VisualCSE-BERT$_{base}$ (CIFAR) | 76.96 |
| VisualCSE-BERT$_{base}$ (ImageNet) | 77.50 |
| SimCSE-RoBERTa$_{base}$ | 76.57 |
| VisualCSE-RoBERTa$_{base}$ (CIFAR) | 77.71 |
| VisualCSE-RoBERTa$_{base}$ (ImageNet) | 77.87 |

Table F.1: Comparison of VisualCSE using CIFAR and ImageNet

# G  Pre-Trained Models & Datasets for Other Languages

For our experiments on German, French, Russian and Chinese, we use the following pre-trained models from *bert-base-german-cased*, *dbmdz/bert-base-french-europeana-cased*, *DeepPavlov/rubert-base-cased*, and *bert-base-chinese*.

The evaluations of SimCSE and VisualCSE for these languages are done on validation sets of the multi-lingual STS-B from HuggingFace, `stsb_multi_mt`.