# OpenReview forum: "Non-Linguistic Supervision for Contrastive Learning of Sentence Embeddings"
_NeurIPS.cc/2022/Conference — NeurIPS 2022 Accept_

### Official Review · Reviewer_RnhL · 2022-07-11

**Rating:** 8
**Confidence:** 4
**Soundness:** 4 excellent
**Presentation:** 4 excellent
**Contribution:** 4 excellent

**Summary:**

This paper presents a contrastive method for learning sentence embeddings that incorporates additional modalities using unaligned data. The method follows the SimCSE approach of clustering similar sentence representations and expands upon it by additionally clustering visual or audio input, named VisualCSE and AudioCSE respectively. Because the approach does not require aligned data between the text and additional modality, the method works for any language and additional modalities can be easily evaluated as well. The authors show that VisualCSE and AudioCSE have clear performance gains in both new approaches across a range of downstream tasks compared to the baseline SimCSE. Additional ablations and experiments also show that the learned sentences embeddings have higher alignment and perform similarly to supervised CSE when training with less data or increasing the level of noise.

**Questions:**

*clarity:*
- In equation's 2 and 3, the loss's superscript wasn't immediately clear to me on the first read and doesn't make sense there to me. Can these just be explicitly mentioned in the text (along the lines of "The unsupervised loss based on data augmentation is as follows" then "For a supervised task like NLI, its loss is defined as:").

*significance:*
- While the authors showed the benefits of VisualCSE on languages other than English (French, German and Russian), I would have liked to see a low-resource lanugage like TODO. This would likely require using another model framework besides BERT.

*questions:*
- In the optimization details section, can you expand on how the number of epochs were chosen for training? It appears to be similar to CSE, though 1 epoch seems quite small.
- Any consideration about intentionally batching images of the same label together to better evaluate the contribution of image labels?
- Seeing how image labels were used for SupCon, is there a correlate experiment in the audio domain to incorporate labels?

*typos:*
- L34: dataset -> datasets
- L43: difference -> different
- L124: representation -> representations
- L218: the improve -> the improvement
- L286: of labeled dataset -> of the labeled dataset
- L345: extend -> extent

*Citations:*
MCSE: Multimodal Contrastive Learning of Sentence Embeddings -> This is essentially concurrent work (put on arXiv in April, presented at NAACL), so it's understandable that it's not cited. I would suggest reading and editing a few small aspects of the paper, e.g. saying this work is the first to look at this approach.

**Limitations:**

yes, limitations are addressed

**Strengths And Weaknesses:**

**Strengths**

*originality:*

- This is a really interesting concept that I'm a big fan of! It's an interesting take on using multimodal input to benefit language learning by using another modality in a completely unaligned way to improve the language embeddings.

*quality:*

- This paper shows a number of experiments analyzing different aspects of the approach -- raw performance on downstream tasks when compared to SimCSE, performance across 2 modalities compared to text-only training, performance across 3 models, performance across different seeds, performance across different languages, and ablations. Because there are so many experiments, it's easy to analyze and think about the contributions of the approach for different communities.
- The visual ablation was interesting as it showed whether supervision from the visual domain, in the form of labels, helps performance. Seeing that it did not contribute much, likely due to the smaller batch size, also leaves room for future work.
- Section 5.1 is another strong ablation, providing a relationship between the number of examples and amount of noise in supervised SimCSE versus unsupervised VisualCSE.
- Experimental setup is thorough, especially by comparing across seeds and therefore across different subsets of the training data.

*clarity:*
- The paper is clearly written and I could follow the motivation, experimental setup and results without really needing additional background searches.

*significance:*
- Not requiring aligned data means virtually any modality can be plugged in here; in addition, the data being used in the existing modalities can easily be expanded or modified. In addition, not being tied to single language is a major contribution! This is especially true when incorporating additional modalities as most work using images is quite limited in the number of languages with annotations.
- For a quantitative perspective, the results in Table 2 show that VisualCSE and AudioCSE nearly always outperform SimCSE across different models and across downstream tasks. These are impressive gains and really validates the idea of the approach.

**Weaknesses**
*clarity:*
- In equation's 2 and 3, the loss's superscript wasn't immediately clear to me on the first read and doesn't make sense there to me. Can these just be explicitly mentioned in the text (along the lines of "The unsupervised loss based on data augmentation is as follows" then "For a supervised task like NLI, its loss is defined as:").

*significance:*
- While the authors showed the benefits of VisualCSE on languages other than English (French, German and Russian), I would have liked to see a low-resource language from outside of Europe.

---

> ### Author Response · Authors · 2022-08-02
> **Response to Reviewer RnhL**
>
>
> We thank the reviewer for their time and their insightful comments and questions. It is really nice to hear that your a fan of the concept presented. We have attempted to address the questions to the best of our ability given the time constraints.
>
> **Q: While the authors showed the benefits of VisualCSE in languages other than English (French, German and Russian), I would have liked to see a low-resource language from outside of Europe.**
>
> Thank you for the suggestion. We rely on STS-B multi-languages for evaluations, which only supports a few popular languages. To show the robustness of our method to non-European languages, we additionally test VisualCSE-bert-base on Chinese, showing that our framework can work on non-European languages as well (see table below).
>
> | Model | STS-B |
> |:---------|:---------:|
> | SimCSE | 67.98 |
> | VisualCSE | 70.05 |
>
> Table 5 of our revised submission includes these new results.
>
>
> **Q: In the optimization details section, can you expand on how the number of epochs were chosen for training? It appears to be similar to CSE, though 1 epoch seems quite small.**
>
> SimCSE trains 1 epoch (15K steps), and we find that actually SimCSE converges even faster (based on validation on STS-B), with a few thousands steps. Thus, training longer in the current setup of SimCSE will not help.
>
> Similarly, we observe that VisualCSE already converges within 1 epoch (15K steps), i.e., reaches best validation on STS-B.
>
>
> **Q: Any consideration about intentionally batching images of the same label together to better evaluate the contribution of image labels?**
>
> This is really a good suggestion to potentially improve the performance under the current hardware limitation. If we see significant improvements, we will include it in our final version.
>
>
> **Q: Seeing how image labels were used for SupCon, is there a correlated experiment in the audio domain to incorporate labels?**
>
> Based on your suggestion, we have run additional loss ablation experiments for the audio domain. The results can be seen in the table below which has been added to Appendix G in our revision. We see a similar trend in audio as we did with images.
>
>
> | Model | STS12 | STS13 | STS14 | STS15 | STS16 | STS-B | SICK-R |  Avg. |
> |:---------|:---------:|:---------:|:---------:|:----------:|:---------:|:---------:|:----------:|:-------:|
> | BERT (SimCLR) | 70.98 | 83.03 | 75.74 | 83.03 | 78.10 | 78.33 | 69.74 | 77.00 |
> | BERT (SupCon) | 71.65 | 84.27 | 76.69 | 83.22 | 78.69 | 79.94 | 70.49 | 77.85 |
> | RoBERTa (SimCLR) | 68.88 | 83.74 | 74.69 | 82.46 | 82.02 | 81.52 | 70.65 | 77.71|
> | RoBERTa (SupCon) | 68.44 | 83.96 | 75.77 | 82.38 | 82.07 | 81.63 | 70.56 | 77.83 |
>
>
> **Q: Citations: MCSE**
>
> We have added a discussion of MCSE in the Related Work of our revision. MCSE is a concurrent work which attempts to learn CSE with images. However, the key difference between MCSE and our framework is that we do not require **aligned** data (e.g., Flickr30k or MSCOCO captioning dataset) for training. The non-reliance on aligned data means that our method can without any additional costs for data curation and annotation learn CSE for different languages and with different modalities (not just images). (As we show in our paper through experiments)
>
> Finally, note that MCSE cannot be run on unpaired data and as such we cannot directly compare it to our method.
>
> We are also grateful to the reviewer for the suggestions on improving the presentation and pointing out typos. We have updated our paper accordingly.

---

### Official Review · Reviewer_sC7n · 2022-07-13

**Rating:** 5
**Confidence:** 4
**Soundness:** 3 good
**Presentation:** 3 good
**Contribution:** 2 fair

**Summary:**

This paper studies the non-linguistic supervision to contrastive-learning-based sentence embedding learning. The author proposes a Transformer-based multi-modal framework to learn text-contrastive and vision/audio-contrastive tasks with unpaired samples. Experiments on various semantic textual similarity tasks show the effectiveness of the proposed method.


**Questions:**

- Could the authors elaborate more on the deficiency of the proposed method in the supervised setting (is this also true for different languages), and could that be alleviated with more unlabeled training data?
- Could the authors elaborate on the epochs used for training the audio and vision model, and will the quality/amount of image/audio affect the performance of the proposed methods?

**Ethics Review Area:**

["I don’t know"]

**Limitations:**

- It would be good for more discussion around the usage (quality and amount) of external multi-modality datasets and will aligned dataset can also be leveraged by the proposed method;


**Strengths And Weaknesses:**

*Summary Of Strengths
- The paper is clearly presented and well-written.
- Extensive empirical experiments with multiple runs not only clearly show the pros/cons for the sts tasks but may suggest the path for future improvement in the low-resource language.
- A simple yet effective modification is proposed and shows consistent improvements over all systems.

*Summary Of Weaknesses
- It seems the paper is missing a comparison with MCSE [1], which is a very relevant work that augments the SimCSE with multimodal contrastive objective;
- It is unclear, except for the objective and hyperparameters, how other factors like pre-trained image/audio encoder, augmentation methods, dataset, and image/audio quality could affect the final performance of the proposed methods;
- It is unclear to me how many epochs/images will the proposed methods improve over the SimCSE methods and how will that affect the final quality;

[1] Zhang, Miaoran, Marius Mosbach, David Ifeoluwa Adelani, Michael A. Hedderich, and Dietrich Klakow. "MCSE: Multimodal Contrastive Learning of Sentence Embeddings." arXiv preprint NAACL (2022).

---

> ### Author Response · Authors · 2022-08-02
> **Response to Reviewer sC7n**
>
> We thank the reviewer for their time and their insightful comments and questions. We have attempted to address the questions to the best of our ability given the time constraints.
>
> **Q: It seems the paper is missing a comparison with MCSE**
>
> Thanks for raising this paper. MCSE was just presented at NAACL this July. However, we have added a discussion of MCSE in the Related Work of our revision. MCSE is a concurrent work which attempts to learn CSE with images. However, the key difference between MCSE and our framework is that we do not require **aligned** data (e.g., Flickr30k or MSCOCO captioning dataset) for training. The non-reliance on aligned data means that our method can without any additional costs for data curation and annotation learn CSE for different languages and with different modalities (not just images). (As we show in our paper through experiments)
>
> Finally, note that MCSE cannot be run on unpaired data and as such we cannot directly compare it to our method.
>
> **Q: It is unclear, except for the objective and hyperparameters, how other factors like pre-trained image/audio encoder, augmentation methods, dataset, and image/audio quality could affect the final performance of the proposed methods;**
>
> We used standard image/audio encoders, standard augmentation from SupCon/SimCLR, and a randomly down-sampled dataset from the commonly used ImageNet/LibriSpeech datasets, showing that our framework with simple naive choices and using readily available tools without modification can improve over SimCSE, thus demonstrating the feasibility of our framework. Relying on default choices reduces the cost of deploying our framework for new modalities and languages. We agree that carefully choosing each part could lead to even stronger results, but that would add additional costs to the deployment of our framework.
>
> **Q: Could the authors elaborate more on the deficiency of the proposed method in the supervised setting (is this also true for different languages), and could that be alleviated with more unlabeled training data?**
>
> Supervised-SimCSE learns from the NLI datasets with positive (entailments) and hard-negative pairs (non-entailments). Our additional supervision from clustering on another modality does not come with such high quality pairs, i.e., we do not have hard negatives for these modalities. The mismatch in the form of losses in text and image/audio may partly explain why the improvements in supervised setting are smaller. In fact, such high quality pairs for text are also only available in English.
>
> We could not evaluate supervised CSE in other languages, because the NLI dataset is not available in other languages. In fact, this is one limitation of supervised CSE, i.e., it relies on a high quality labeled dataset for training which is often not available in other languages.
>
> Since we think that the main problem is the lack of high quality positive / hard-negative pairs in other modalities we don’t believe that more unlabeled training data can further improve the supervised setting. Also, we find that the training of SimCSE and our VisualCSE usually converge (based on validation loss) in a few thousands steps, meaning that having more unlabeled training data does not improve the performance of our current framework. One possibility to leverage additional data is to increase the image batch size (then within the same number of steps, more images are leveraged). However, due to hardware limitations (GPU memory limitations), we set the batch size to be 48 in all our experiments.
>
> **Q: Could the authors elaborate on the epochs used for training the audio and vision model, and will the quality/amount of image/audio affect the performance of the proposed methods?**
>
> unsup-SimCSE trains for one epoch, and converges in thousands of steps. Thus, we also simply match the training policy of SimCSE and train for only one epoch. The number of images used for training (30K) is chosen such that during the training until convergence the model will iterate through the image dataset a few times (when using a batch size of 48).
>
> We further carry out experiments on substituting ImageNet with a lower quality CIFAR for VisualCSE. CIFAR images have a shape of 32x32 and we intentionally resize (enlarge) them to 224x224 to be encoded by the ViT embedding layer. This interpolation causes the CIFAR images to become blurry and lower quality. Our experiments show that our framework improves over SimCSE even with this lower quality dataset (results shown below).
>
> RoBERTa-base-uncased:
> | Model | Avg. |
> |:--------|:-----:|
> | SimCSE | 76.57 |
> | VisualCSE (CIFAR) | 77.71 |
> | VisualCSE (ImageNet) | 77.87 |
>
>
> BERT-base-uncased:
> | Model | Avg. |
> |:--------|:-----:|
> | SimCSE | 76.25 |
> | VisualCSE (CIFAR) | 76.96 |
> | VisualCSE (ImageNet) | 77.50 |
>
>
> We have added these two tables in Appendix H of our revision.

---

> > ### Comment · Reviewer_sC7n · 2022-08-08
> > **Reply to Authors**
> >
> > The detailed response from the authors solved most of my concerns. After reading the author's response and other review comments, I decided to keep my original rating as "Borderline Accept."

---

### Official Review · Reviewer_TGns · 2022-07-15

**Rating:** 4
**Confidence:** 4
**Soundness:** 3 good
**Presentation:** 3 good
**Contribution:** 2 fair

**Summary:**

This work is an extension of existing contrastive sentence representation learning. Instead of only using text data, this work further bring the other modality to improve the model performance. In this work, the author proposed unpaired image or audio data as additional supervision signals. The extra modality data share the same encoding backbone models and similar contrastive learning losses. The author conducted experiments on standard sentence similarity evaluation datasets, and achieved improvements over text-only models.

**Questions:**

1. Is it possible to use same CL losses for different modalities?
2. What if we use both image and audio data together? Can if further improve the perforamnce?
3. Is there token-level alignment between different modals?
4. If the proposed mode can further improve the image-text alignment compared with our models, e.g., CLIP?

**Limitations:**

I do not see important social limitations of this work.

**Strengths And Weaknesses:**

Strengths: The proposed idea is simple and reasonable. By introducing image/audio through the same transformer encoder, the model transferred from text only to modal agnostic model. This is quite different from other works which use separated structures for each modality. One good point of the model is that the image data size (500) is much smaller than the text training data. This means the model extra cost is small. The author also well analyzed the proposed model.

Weakness: The model performance improvement is relatively small. Compared to other CL sentence representation learning work, e.g., DiffCSE, DebiasedCSE, PromptBERT, the proposed method performance with more data is not that significant. The author also did not evaluate the performance on other modalities, then it is hard to give a overall judgment of the model as a modal agnostic mode. The overall training loss on different modality losses are also missing which in my view in also important to understand the contribution of different modalities.

---

> ### Author Response · Authors · 2022-08-02
> **Response to Reviewer TGns**
>
> We thank the reviewer for their time and their insightful comments and questions. We have attempted to address the questions to the best of our ability given the time constraints.
>
> **Q: Compared to other CL sentence representation learning work, e.g., DiffCSE, DebiasedCSE, PromptBERT, the proposed method performance with more data is not that significant.**
>
> Thanks for pointing out these related works, we have added them to our discussion of related work in the revision. Note that these methods mentioned are all extensions of SimCSE, in that they all rely on a unimodal text dataset. For instance, DiffCSE [1] can be viewed as SimCSE *by equivariant loss*. Whereas, in our paper, we show a framework for combining text-based contrastive learning (SimCSE) with contrastive learning in other modalities (images or audio). Technically, our framework complements text-based contrastive learners (such as SimCSE and its variants) with non-linguistic contrastive learners. This means that other variants of SimCSE, such as DiffCSE should see improvements in our framework. To test this, we carried out further experiments with DiffCSE + image clustering loss using ImageNet using BERT (base-uncased). Our experiments show that using our framework with the ImageNet dataset, DiffCSE can on average be improved by 1.0% absolute for the STS tasks (comparable to the average improvement of 1.25% absolute when using SimCSE in our framework). This shows that our framework, which clusters examples independently from two different modalities, is agnostic to the choice of the text base learner (e.g., SimCSE or DiffCSE) and can be used to further improve SimCSE variants such as DiffCSE.  Due to time limitations we were not able to run experiments with the audio data, but we have no reason to believe that similar results should not be expected there as well (we will add these experiments to our camera ready when the audio experiments finish).
>
> [1]Yung-Sung Chuang, Rumen Dangovski, Hongyin Luo, Yang Zhang, Shiyu Chang, Marin Soljačić, Shang-Wen Li, Wen-tau Yih, Yoon Kim, James Glass, DiffCSE: Difference-based Contrastive Learning for Sentence Embeddings, NAACL 2022
>
> **Q: Is it possible to use the same CL losses for different modalities?**
>
> Yes, it is possible as the loss is based on the nature of the task (i.e., contrastive loss) and not the modality of the dataset. For instance, we use the exact same loss for both the image and audio modalities in our paper.
>
>
> **Q: What if we use both image and audio data together? Can it further improve the performance?**
>
> Our loss can be viewed as a multi-task loss, with each modality being analogue to a different task. As prior work has shown, in multi-task settings naively adding additional tasks does not necessarily improve the performance of models [1]. In fact, recent work by Liu et al. [2] shows that even with only three tasks, effective multitask learning requires a meta-learning mechanism. However, this does not mean that going beyond one non-linguistic modality cannot improve the performance, it only insinuates that it is not guaranteed. We have initiated experiments  to find out whether audio+image together can improve the performance of our framework. However, the joint loss makes the hyper-parameter search space grow combinatorially and as such additional time is required for our experiments to finish. We hope to report the results in our camera ready.
>
> [1] Trevor Standley, et al. Which Tasks Should Be Learned Together in Multi-Task Learning? ICML 2020
>
> [2] Liu, Shikun, et al. "Auto-Lambda: Disentangling Dynamic Task Relationships. TMLR 2022
>
> **Q: Is there token-level alignment between different modals?**
>
> No, our learning algorithm does not require or assume such alignment.  Batches for text and images (or audio) are randomly chosen at each step. We do not do further alignment for text and other modals in each training step.
>
> **Q: If the proposed mode can further improve the image-text alignment compared with other models, e.g., CLIP?**
>
> Our framework is not designed for alignment of text and other modals (such as image). Our framework is for learning better sentence representations using non-aligned/non-parallel datasets of text and other modals (evaluated using STS, the standard sentence embedding benchmark). Generalization to other tasks like improving image-text alignment with **unpaired** images and text examples could be a very interesting future work, however, it is outside the scope of this paper.

---

### Official Review · Reviewer_tnhg · 2022-07-16

**Rating:** 6
**Confidence:** 3
**Soundness:** 3 good
**Presentation:** 3 good
**Contribution:** 3 good

**Summary:**

This work drives a plausible approach for enhancing unsupervised sentence embedding learning by leveraging unpaired examples from another modality. Motivated by the previous success of Lu et al. [33], Transformer models can transfer knowledge by learning from another modality, e.g. vision or audio. The text part uses the same training paradigm as SimCSE, while an additional image contrastive loss is incorporated by using the same language transformer for image encoding. Results show that the proposed method is language-agnostic and outperforms vanilla SimCSE across various tasks. Analysis versus supervised SimCSE also illustrated that the proposed approach performs better than SimCSE learned with a noisy and smaller NLI dataset.

**Questions:**

Please refer to weaknesses for score-related questions.

- Can the supervision on non-linguistic dataset (e.g. image labels) further enhance the proposed model?

**Limitations:**

The authors discussed limitations on the reduction in Uniformity of the sentence embeddings. Other possible limitations could be the marginal improvement with larger non-linguisitic dataset.

**Strengths And Weaknesses:**

Strengths:
- The idea of using non-linguistic supervision for sentence embedding learning is novel.
- The proposed model does not require paired multi-modal data for training, which can be applied to low-resource language learning.
- The finding that Transformer models can generalize by learning a similar task across different modalities may shed light on future research on multi-modal representation learning.
- The experiments demonstrate its success on both VisualCSE and AudioCSE.

Weaknesses:
- Though the authors attempted to answer the underlying rationale of the proposed model in Sec. 3.4, how the language model can be trained using other modalities remains unclear to me. More analysis experiments are expected to strengthen the theoretical support for the framework.
- The work has discussed how language dataset supervision can affect the final performance. However, the non-linguistic dataset quality is not discussed. What kind of non-linguistic dataset is suitable for the paradigm? Does ImageNet fit for all language training?

Minors:
- Line 172: "thesame" $\rightarrow$ "the same"

---

> ### Author Response · Authors · 2022-08-02
> **Response to Reviewer tnhg**
>
> We thank the reviewer for their time and their insightful comments and questions. We have attempted to address the questions to the best of our ability given the time constraints.
>
> **Q: How the language model can be trained using other modalities remains unclear to me., More analysis experiments are expected to strengthen the theoretical support for the framework.**
>
> The intuition behind our framework comes from recent work by Lu et al. [1] that hypothesize that language pre-trained transformers (such as the ones used in our paper) can act as “universal computational engines”, meaning that they share knowledge between text and other modalities. In their experiments, Lu et al. already show that this phenomenon is unique to transformers (e.g., compared to LSTMs). They also investigate several other factors (such as model size) to further strengthen their hypothesis (see section 3 in their paper).
>
> Though our framework is different (as it relies on a multi-task loss without parallel data), our findings provide further support for the “universal computational engines” hypothesis for at least two non-text modalities (vision and audio). In our experiments, we show that Transformers indeed do become better contrastive learners by jointly learning with similar contrastive objectives in different modalities.
>
> [1] Kevin Lu, Aditya Grover, Pieter Abbeel, and Igor Mordatch. Pretrained transformers as universal computation engines, AAAI-2022.
>
> **Q: What kind of non-linguistic dataset is suitable for the paradigm? Does ImageNet fit for all language training?**
>
> Because contrastive sentence embedding learning is about *clustering* and because the data needs to be encoded before being passed to the transformer model, the requirements for a non-linguistic dataset to be used for the paradigm are that it should be suitable for clustering (i.e., be labeled) and that a suitable embedding layer exists for that data. We used ImageNet and LibriSpeech for our non-linguistic datasets as they are commonly used and easily accessible labeled datasets. For both datasets we used around 30K samples. It is possible that careful cherry picking of the non-linguistic dataset could lead to improved performance, but that was not the objective of this paper. The advantage of not cherry picking the non-linguistic datasets is that our framework can be easily applied without much additional cost. As we show in Section 4.5, ImageNet was fit for training in all languages that we experimented with (including a newly added experiment on Chinese, showing that our framework can work on non-European languages as well). In brief, our experiments show that ImageNet should suffice as the non-linguistic dataset for all language training.
>
> **Q: Can the supervision on non-linguistic dataset (e.g. image labels) further enhance the proposed model?**
>
> This is a good question which we were curious about as well. We experiment with both SupCon, which is a contrastive loss that leverages labels, and SimCLR, which is a loss that does not leverage labels for both the image and audio datasets. As we show in Table 4,for images, a loss that leverages the labels (i.e., SupCon) has a tiny advantage over a loss that does not (i.e., SimCLR). This is possibly due to the fact that our image batch size of 48 is not large enough. As we discuss in the paper: *As we have 60 image classes, each with 500 images, in our down-size ImageNet dataset used for training, it is not likely that multiple examples from the same class are contained in one mini-batch*. Thus, due to our current hardware limitations, we are unable to harness more from the image labels. We report similar results for the audio dataset in Appendix G.

---

### Official Review · Reviewer_uUn7 · 2022-07-22

**Rating:** 6
**Confidence:** 3
**Soundness:** 2 fair
**Presentation:** 2 fair
**Contribution:** 2 fair

**Summary:**

Summary:
* This paper proposes a novel contrastive learning method of sentence embeddings, using unpaired examples from another modality. Experiments on semantic textual similarity benchmarks demonstrate that this method is able to produce sentence embeddings higher quality.


**Questions:**

* Did you do hyper-parameters search for both SimCSE baseline and VisualCSE method?
* Have you tried to compare to SimCSE with “supervised models” setting?
* What’s the intuition behind sharing everything else besides the embedding layer for image and sentence encoders?
* Have you tried tasks besides STS?



**Strengths And Weaknesses:**

Strength:
* This paper presents a novel method to conduct multimodal contrastive learning for sentence encoders.
* This method is simple and generalizable, i.e. not only applied to vision but also to other modalities such as audio.

Weakness:
* I’m not confident that the improvement is significant. On STS tasks, on average it improves 1% accuracy.  In comparison, the original SimCSE paper improves more than 4% accuracy and is able to push the state-of-the-art in a supervised setting.  I wonder how much variance these datasets have, and how much hyper-parameters sweeps is conducted for both methods.
* The model setup is a bit unintuitive. In order to conduct multimodal contrastive learning, it assumes that we can replace the embedding layer of a sentence encoder with a visual embedding layer to get an image encoder.

---

> ### Author Response · Authors · 2022-08-02
> **Response to Reviewer uUn7**
>
> We thank the reviewer for their time and their insightful comments and questions. We have attempted to address the questions to the best of our ability given the time constraints.
>
> **Q: I’m not confident that the improvement is significant.**
>
> Though we agree that the performance boost is not great, we show that the boost is consistent by running repeated experiments with different random seeds (see section 4.3) and report the standard deviations.
>
> However, note that our framework is language and modality agnostic, and that it does not require any paired examples across modalities. This means that our framework can be used with very little additional costs across a wide range of languages (we included new experiments on Chinese in Section 4.5 to show our method’s applicability to non-European languages).   Furthermore, by thorough experimentation, our paper demonstrates the capability of Transformers to generalize better by learning a similar task with multi-task losses, even without parallel examples. Other than practical implications, these findings have conceptual significance as they support the hypothesis that Transformer models can share knowledge between different modalities.
>
> **Q: Did you do hyper-parameters search for both SimCSE baseline and VisualCSE method?**
>
> Yes, for SimCSE we searched through {64,128,256,512}, {1e-5, 2e-5...5e-5} and closely reproduced the results reported in the SimCSE paper. Similarly, we searched for hyper-parameters for VisualCSE/AudioCSE (see Section 4.1 and Appendix C).
>
> **Q: Have you tried to compare SimCSE with “supervised models” setting?**
>
> We did this in Section 4.6 and results are shown in Table 6.  As mentioned in the paper: *We observe that Visual/AudioCSE can still outperform supervised SimCSE in all the tasks, but with much smaller margins. This shows that the learning signal from the high-quality negative and positive pairs of the NLI dataset are very strong (leading to a 5.32 improvement over unsupervised SimCSE) and cannot be supplemented by supervision from other modalities.''*
>
> **Q: What’s the intuition behind sharing everything else besides the embedding layer for image and sentence encoders?**
>
> Our intuition is *inspired by Lu et al. [1] which shows the ability of Transformers to transfer knowledge between text and other modalities (e.g., they show that a Transformer model pre-trained on text can be fine-tuned on downstream visual tasks).*
>
> Note that Lu et al. [1]  investigate knowledge sharing between text and other modalities through *transfer learning*, i.e They do not introduce additional parameters and freeze most parameters learned from language and fine-tune a few layers (e.g., normalization) on downstream vision tasks. Our framework shows that knowledge can be shared between text and other modalities in a *multi-task setting*. Other than practical implications, these findings have conceptual significance as they support the hypothesis that Transformer models can share knowledge between different modalities.
>
> [1] Kevin Lu, Aditya Grover, Pieter Abbeel, and Igor Mordatch. Pretrained transformers as universal computation engines, AAAI-2022
>
> **Q: Have you tried tasks besides STS?**
>
> As discussed in SimCSE, the main goal of sentence embedding learning is to generate better semantic representations for sentences, which can be directly evaluated through the STS task.
>
> However (following SimCSE’s Appendix), we also evaluated our models in a few downstream classification tasks (Mr, Cr, Subj, MPQA, SST-2, TREC and MRPC) in Appendix E. Note that these tasks are not the main goal of sentence embedding learning, but we (VisualCSE and AudioCSE) still get consistent improvements over SimCSE.

---

### Author Response · Authors · 2022-08-02
**General Response**

We sincerely thank all the reviewers for their time and their thoughtful comments and questions. We are encouraged that the reviewers find our method novel (uUn7, Tnhg, RnhL), simple yet effective (uUn7, TGns, sC7n) and generalizable (uUn7, Tnhg), and that the paper is well-presented (sC7n, RnhL) and contains extensive experiments (sC7n, RnhL).

We attempted our best to address the questions as time allowed and have revised the paper accordingly. We believe the revisions have made the paper stronger and thank all the reviewers for their help. Please find individual responses to your questions below.

---

### Meta-Review · Area_Chair_RqxZ · 2022-08-28

**Recommendation:** Accept
**Confidence:** Certain

**Metareview:**

This paper improves contrastive learning of sentence embedding by using unpaired examples from the image or audio modality.

Reviewers liked the significance of this work due to its simplicity and general applicability, but some questioned the amount of improvement and advocated for the inclusion of low resource languages. The authors included Chinese, which is non-European but not low resource.

**Award:**

No

---

### Decision · Program_Chairs · 2022-09-14

Accept